# Dysregulation of the NUDT7-PGAM1 axis is responsible for chondrocyte death during osteoarthritis pathogenesis

Jinsoo Song[1], In-Jeoung Baek[2], Churl-Hong Chun[3] & Eun-Jung Jin [1]

Osteoarthritis (OA) is the most common degenerative joint disease; however, its etio-pathogenesis is not completely understood. Here we show a role for *NUDT7* in OA patho-genesis. Knockdown of *NUDT7* in normal human chondrocytes results in the disruption of lipid homeostasis. Moreover, *Nudt7*$^{-/-}$ mice display significant accumulation of lipids via peroxisomal dysfunction, upregulation of *IL-1β* expression, and stimulation of apoptotic death of chondrocytes. Our genome-wide analysis reveals that *NUDT7* knockout affects the gly-colytic pathway, and we identify *Pgam1* as a significantly altered gene. Consistent with the results obtained on the suppression of *NUDT7*, overexpression of *PGAM1* in chondrocytes induces the accumulation of lipids, upregulation of *IL-1β* expression, and apoptotic cell death. Furthermore, these negative actions of *PGAM1* in maintaining cartilage homeostasis are reversed by the co-introduction of *NUDT7*. Our results suggest that *NUDT7* could be a potential therapeutic target for controlling cartilage-degrading disorders.

[1] Department of Biological Sciences, College of Natural Sciences, Wonkwang University, Iksan, Chunbuk 54538, Republic of Korea. [2] Asan Institute for Life Sciences, University of Ulsan College of Medicine, Seoul 05505, Republic of Korea. [3] Department of Orthopedic Surgery, Wonkwang University School of Medicine, Iksan, Chunbuk 54538, Republic of Korea. Correspondence and requests for materials should be addressed to E.-J.J. (email: jineunjung@wku.ac.kr)

Osteoarthritis (OA), a major health and economic burden on the elderly population worldwide, is a degenerative joint disorder characterized by the gradual degradation of articular cartilage and synovial inflammation[1]. The chondrocyte, the unique resident cell type in articular cartilage, regulates the anabolic and catabolic pathways to maintain cartilage homeostasis[2,3]. Activation of catabolic regulators leads to the release of matrix metalloproteinases (MMPs), disintegrin, and metalloproteinases with thrombospondin motifs (ADAMTS), which eventually contribute to the disruption of cartilage homeostasis and apoptosis of chondrocytes[4,5]. Although many multifactorial risk factors contributing to the imbalance between catabolic and anabolic pathways during OA pathogenesis have been identified, the crucial molecular regulators and underlying regulatory mechanism remain unknown.

Recently, our laboratory showed that impaired peroxisomal function is closely involved in OA pathogenesis[6,7]. Peroxisomes are known to be vital for various metabolic pathways, such as β-oxidation of very long chain fatty acid (VLCFA), ether lipid synthesis, bile acid metabolism, and reactive oxygen species (ROS) metabolism[8–10]. Peroxisomal disorders could occur as the result of defects in the biogenesis of peroxisomes or in peroxisomal enzymes[11,12]. Nudix hydrolases, which are peroxisomal enzymes, function to remove toxic nucleotide metabolites from the cell and regulate the concentration of many different nucleotide substrates, cofactors, and signaling molecules[13,14]. Among nudix hydrolases, enzyme family member 7, NUDT7, acts as a coenzyme A (CoA) diphosphatase, which mediates the cleavage of CoA[15,16]. NUDT7 functions as a house-keeping enzyme by eliminating potentially toxic nucleotide metabolites, such as oxidized CoA from β-oxidation in the peroxisome, as well as nucleotide diphosphate derivatives, including $NAD^+$, NADH, and ADP-ribose[17–19]. NUDT7 expression is high in the liver and low in the brain and heart[15]. Downregulation of NUDT7 in mice accelerated senescence[20], and suppressed expression level was observed in the liver of starved mice[21]. Despite the important role of NUDT7 in cellular metabolism and function, only few studies have shown the possible association between NUDT7 and pathobiology, and these studies were limited to plants. NUDT7 knockout ($Nudt7^{-/-}$) in Arabidopsis induced growth retardation by the reduction in the ATP level caused by an alteration of the $NAD^+$/NADH ratio[17]. Arabidopsis thaliana NUDT7 is also reported as a negative regulator of the enhanced disease susceptibility 1 (EDS1)-mediated defense response[22,23]. The loss of NUDT7 function in Arabidopsis enhanced resistance to virulence and elevated sensitivity to oxidative stress[17,24]. To date, the link between NUDT7 and OA pathogenesis has not been investigated, even though the cellular and biological functions of NUDT7 could be closely related to OA pathogenesis. In the present study, we aimed to investigate the functional role and regulatory mechanism of NUDT7 during OA pathogenesis. The results suggested that NUDT7 is a positive regulator in maintaining cartilage homeostasis.

## Results

### NUDT7 is involved in OA pathogenesis.
Cartilages of patients who undergo total knee replacement (TKR) surgery were divided into a relatively healthy area (designed as Non-OA) and a severely damaged area (designed as OA), and the histological characteristics of OA were confirmed by analyzing the degradation of the cartilage matrix (Fig. 1a). In the OA cartilage, markedly less staining with safranin O and intensive staining of markers reflecting cartilage degradation, such as Collagen C1-2C, Aggrecan Neoepitope, and matrix metalloproteinase (MMP)-13, were observed.

Previously, our laboratory suggested the involvement of peroxisomal dysfunction in OA pathogenesis[6,7]. Consistent with

the results in our previous report[6], the activities of glutathione peroxidase (GPx), catalase, and β-oxidation (Supplementary Fig. 1b-d), as well as the expression levels of cartilage matrix genes, such as COL2A1 (collagen type 2 A1), COMP (cartilage oligomeric matrix protein), and ACAN (aggrecan), were dramatically suppressed (Supplementary Fig. 1e), whereas the concentration of free fatty acid was significantly increased in OA chondrocytes (Supplementary Fig. 1f). To identify the responsible peroxisomal genes involved in OA pathogenesis, we analyzed the expression levels of 94 peroxisomal genes taken from the peroxisome database (http://www.peroxisomedb.org) and gene ontology-based database (http://amigo.geneontology.org) (Fig. 1b). Among them, NUDT7 was identified as a gene whose expression was dramatically altered in OA chondrocytes compared with that in non-OA chondrocytes. The transcription and translation of NUDT7 was also significantly downregulated in OA chondrocytes compared with that in normal chondrocytes (Fig. 1c)

To investigate the role and function of NUDT7 in OA pathogenesis, the expression level of NUDT7 was modulated exogenously using a short hairpin RNA (shRNA) (Fig. 1d–i). By knockdown of NUDT7 (shNUDT7), the expression level of COL2A1 was significantly decreased, whereas the expression of cartilage-degrading enzymes, such as MMP13, and ADAMTS-4 and -5 was significantly increased (Fig. 1e). Significant increases in chondrocyte apoptosis and lipid accumulation (Fig. 1g and Supplementary Fig. 2) were also observed in chondrocytes containing shNudt7. In contrast, chondrocyte proliferation (Fig. 1f), β-oxidation and peroxisome numbers (Supplementary Fig. 2), and the activities of GPx and catalase were significantly decreased by the introduction of shNudt7 (Fig. 1h and i). These negative effects on cartilage homeostasis were then reversed by expressing NUDT7 in OA chondrocytes.

### $Nudt7^{-/-}$ mice display the typical OA characteristics.
To investigate the functional role of NUDT7 in vivo, we generated NUDT7 knockout mice ($Nudt7^{-/-}$) using transcription activator-like effector nucleases (TALEN) technology (Fig. 2). Severe cartilage degradation, as assessed by safranin O staining, was observed in the cartilage of destabilization of the medial meniscus (DMM)-induced $Nudt7^{-/-}$ mice (Fig. 2a). In addition, a dramatic reduction in cartilage thickness was observed in DMM-induced $Nudt7^{-/-}$ mice compared with control mice ($Nudt7^{+/+}$, Fig. 2b). Severe cartilage degradation was also confirmed by staining for Collagen C1-2C and Aggrecan Neoepitope in the cartilage of DMM-induced $Nudt7^{-/-}$ mice. The number of Collagen C1-2C-positive cells increased up to 2.5-fold in the cartilage of DMM-induced $Nudt7^{-/-}$ mice compared with that in the cartilage of $Nudt7^{+/+}$ mice (Fig. 2c). Moreover, the chondrocyte numbers in the cartilage were significantly decreased in DMM-induced $Nudt7^{-/-}$ mice (Fig. 2b).

Phytol, a metabolic precursor of phytanic acid, is known to affect peroxisomal proliferation and function through regulation of fatty acid metabolism[25,26]. In mice fed a phytol (Phytol enriched diet, PED), which is under peroxisomal functional stress, severely deteriorative OA phenotypes, i.e., increased cartilage degradation, reduced cartilage thickness, and reduced chondrocyte numbers, were observed in DMM-induced $Nudt7^{-/-}$ mice compared with those in $Nudt7^{+/+}$ mice (Fig. 2a–c). Severe cartilage degradation was also induced by PED without DMM surgery (Supplementary Fig. 3).

### Nudt7 KO induces peroxisomal dysfunction and apoptosis of iMACs.
To explore the role of NUDT7 in chondrocyte differentiation, immature murine articular chondrocytes (iMACs) from

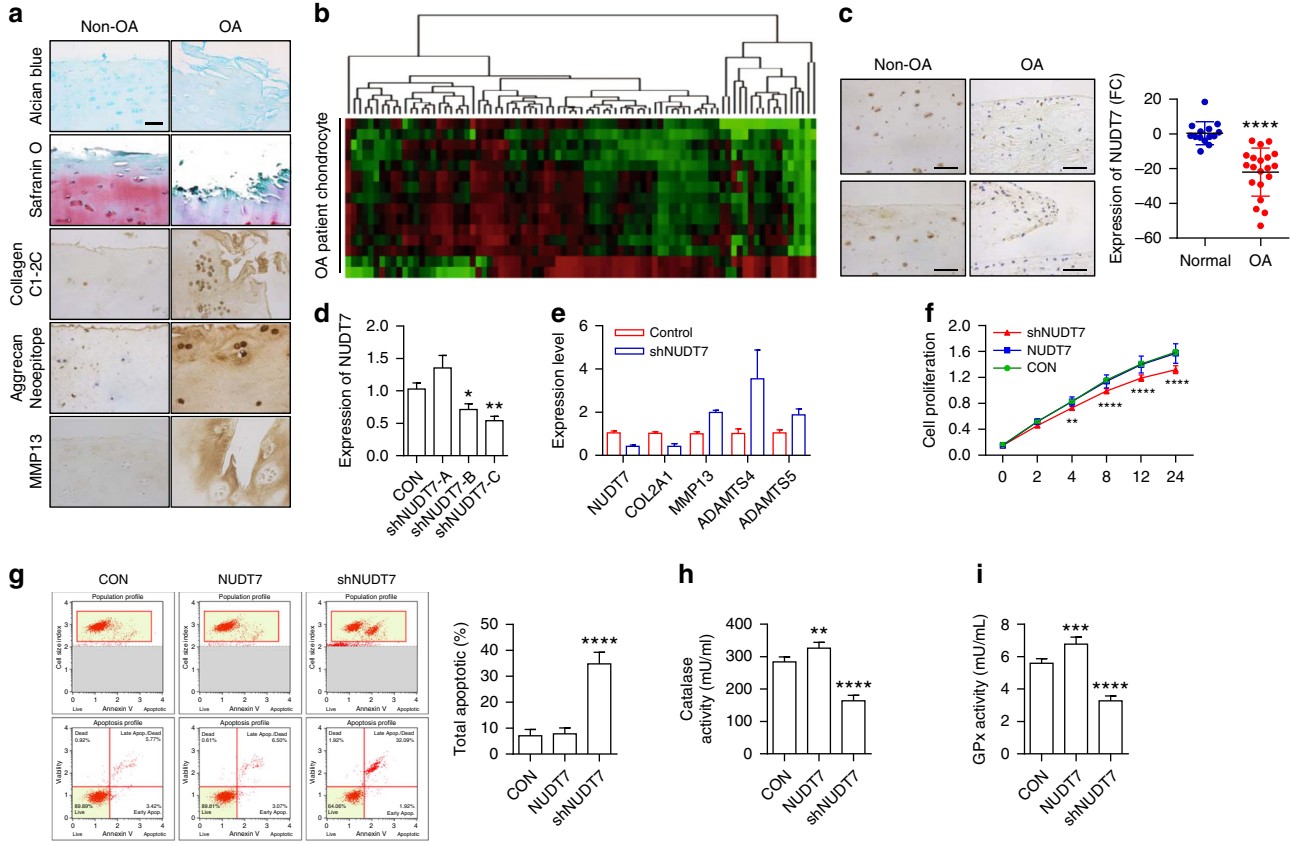

**Fig. 1** *NUDT7*, a peroxisomal gene, is suppressed during osteoarthritis (OA) pathogenesis. **a** Staining with Alcian blue or safranin O and immunohistochemistry of Collagen C1-2C, Aggrecan Neoepitope, and matrix metalloproteinase (MMP)-13 using non-OA and OA cartilage from human patients with OA pathogenesis ($n = 5$ per group). Scale bar, 50 μm. **b** Transcription levels of 94 peroxisome-related genes in human OA chondrocytes ($n = 15$ per group) compared with those in normal chondrocytes ($n = 8$ per group). **c** Immunostaining of NUDT7, and the mRNA level of *NUDT7* in human OA chondrocytes ($n = 20$ per group) compared with that in normal chondrocytes ($n = 13$ per group). Scale bar, 50 μm. **d** Transcription level of *NUDT7* in normal chondrocytes infected with lentivirus containing each *NUDT7* short hairpin RNA (shRNA). **e** Transcription level of matrix-degrading enzymes in normal chondrocytes infected with lentivirus containing an *NUDT7* shRNA (*shNUDT7*) ($n = 3$). **f**, **g** Analysis of cell proliferation ($n = 4$ per group) and apoptotic cell death ($n = 6$ per group) in normal chondrocytes infected with lentivirus containing *NUDT7* or *NUDT7* shRNA. **h**, **i** Activities of catalase and glutathione peroxidase (GPx) in normal chondrocytes infected with lentivirus containing *NUDT7* or *NUDT7* shRNA ($n = 6$ per group). Values are means + s.d. An unpaired Student's *t* test was used for statistical analysis. *$P < 0.05$, **$P < 0.01$, ***$P < 0.001$, ****$P < 0.0001$

*Nudt7*[-/-] mice were cultured with/without a phytol-enriched environment (Fig. 2d–g). The number of Alcian blue-positive cells was significantly decreased, whereas that of BODIPY-positive cells significantly increased among iMACs from *Nudt7*[-/-] mice compared with iMACs from *Nudt7*[+/+] mice (Fig. 2d, e). Moreover, the transcript levels of *COL2A1*, chondrocyte proliferation, and the activities of catalase and GPx were significantly decreased (Supplementary Fig. 4), whereas expression of cartilage-degrading enzymes, such as *MMP-13* and *-9*, *ADAMTS-4* and *-5* (Fig. 2g), and chondrocyte apoptosis (Fig. 2f) were significantly increased in iMACs from *Nudt7*[-/-] mice. Under phytol-enriched conditions, mitochondrial dysfunction and pyruvate accumulation were observed in iMACs from *Nudt7*[-/-] mice (Supplementary Fig. 5a,b). Moreover, an increase in the number of fatty acid synthase (FASN)-positive cells was also observed in human OA cartilage compared with that in non-OA cartilage (Supplementary Fig. 5c).

**NUDT7 affects the glycolytic pathway by regulating PGAM1.**
To identify the regulatory factors in cartilage homeostasis that are modulated by *NUDT7*, we extracted overlapping genes from RNA sequencing of human OA chondrocytes and murine *Nudt7*[-/-] chondrocytes, and performed Ingenuity Pathway Analysis (IPA) (Fig. 3a). IPA suggested that gluconeogenesis, glycolysis, Rapoport-Luebering glycolytic shunt, and N-acetylglucosamine degradation are involved (Fig. 3a), and gene set enrichment analysis (GSEA) showed that glycolysis and fatty acid metabolism are common pathways in human OA chondrocytes and murine *Nudt7*[-/-] chondrocytes (Supplementary Fig. 6). The gene encoding glycolytic enzyme phosphoglycerate mutase (*PGAM1*), a catalytic enzyme that converts 3-phosphoglycerate (3-PG) to 2-phosphoglycerate (2-PG) during glycolysis[27], was one of the altered genes whose expression was most significantly increased in both human OA chondrocytes and murine *Nudt7*[-/-] chondrocytes. Significant upregulation of *PGAM1* in human OA chondrocytes (normal: $n = 8$, average age = 43.84; OA: $n = 15$, average age = 65.55) was re-confirmed by reverse transcription polymerase chain reaction (RT-PCR) (Fig. 3b). Immunohistochemical analysis using human OA cartilage and cartilage of DMM-induced *Nudt7*[-/-] mice showed a significant increase in the expression level of *PGAM1* (Fig. 3c,d). We also observed the accumulation of 2-PG, acetyl-CoA, and malonyl-CoA in iMACs of *Nudt7*[-/-] mice (Fig. 3e–g).

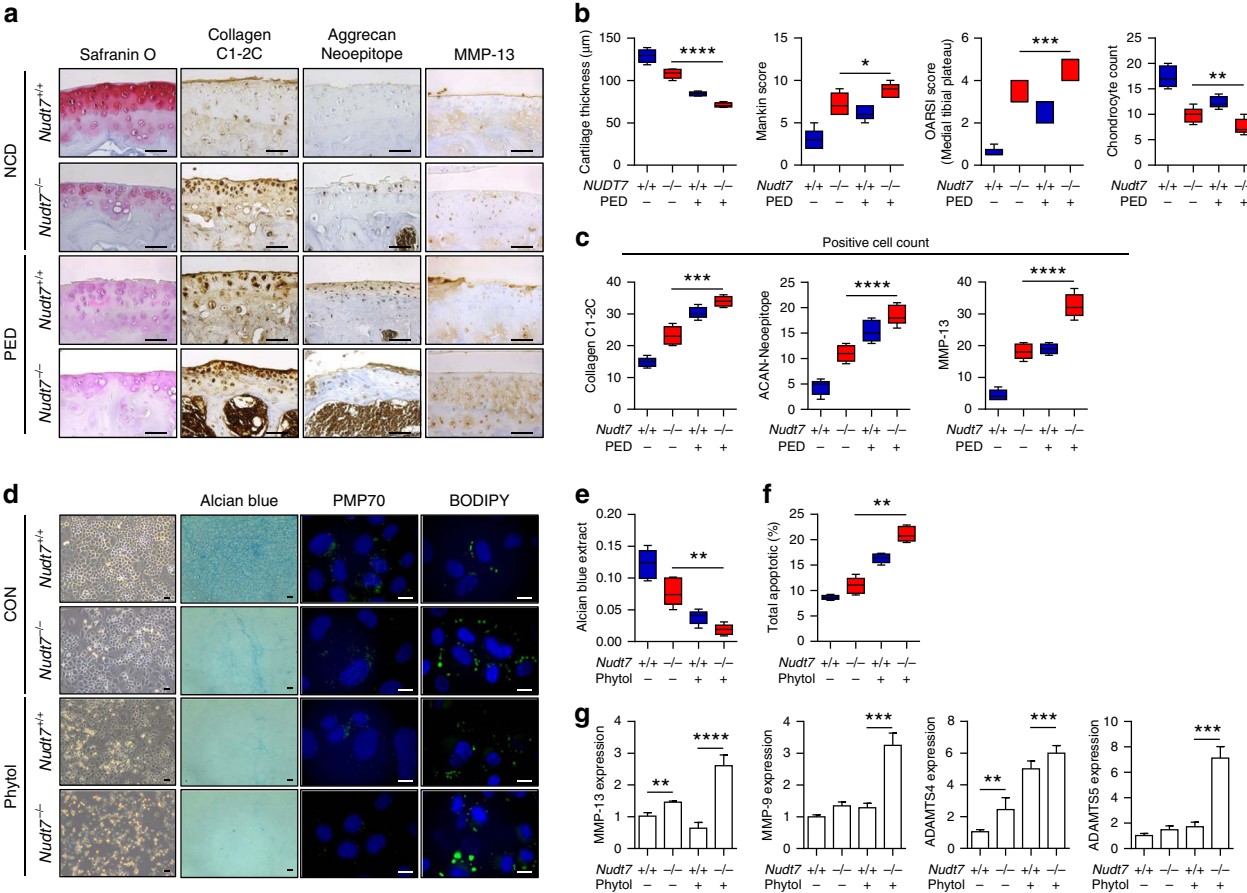

**Fig. 2** *Nudt7*$^{-/-}$ mice have severe cartilage degradation and chondrocyte degeneration. **a** Staining of Collagen C1-2C, Aggrecan Neoepitope, and matrix metalloproteinase (MMP)-13 of paraffin sections from the articular cartilage of *Nudt7*$^{+/+}$ and *Nudt7*$^{-/-}$ mice (*n* = 5 per group). Scale bars, 50 μm. **b** Analysis of cartilage thickness, Mankin score, and chondrocyte number using destabilization of the medial meniscus (DMM)-induced *Nudt7*$^{+/+}$ or *Nudt7*$^{-/-}$ mice under a normal- (NCD) or phytol-enriched diet (PED) from Fig. 2a (*n* = 5 per group). **c** Positive cell number of Collagen C1-2C, Aggrecan (ACAN) Neoepitope, and MMP-13 in DMM-induced *Nudt7*$^{+/+}$ or *Nudt7*$^{-/-}$ mice under a NCD or PED from Fig. 2a (*n* = 5 per group). **d** Staining of Alcian blue, PMP70, BODIPY$^{493/508}$ using iMACs of *Nudt7*$^{+/+}$ and *Nudt7*$^{-/-}$ mice at postnatal day 6 under a control (CON) or 50 mM phytol treatment (phytol; *n* = 5 per group). Scale bars, 10 μm. **e** Quantification of Alcian blue staining extracted with 6M Guanidine-HCl was measured in 650 nm absorbance (*n* = 5 per group). **f** Analysis of apoptotic cell death in iMACs of *Nudt7*$^{-/-}$ mice at postnatal day 6 under CON or phytol conditions compared with those of *Nudt7*$^{+/+}$ mice (*n* = 6 per group). **g** Transcription level of MMP-13 and -9, ADAMTS-4 and -5 in CON and phytol in iMACs of *Nudt7*$^{-/-}$ mice at postnatal day 6 under CON or phytol conditions compared with those of *Nudt7*$^{+/+}$ mice (*n* = 3 per group). Values are means + s.d. A one-way ANOVA was used for statistical analysis (**b**, **c**, **e** and **f**). Values are means + s.d. An unpaired Student's *t* test was used for statistical analysis (**g**). *P < 0.05, **P < 0.01, ***P < 0.001, ****P < 0.0001

**PGAM1 activates inflammatory cytokines and chondrocyte apoptosis.** Inflammatory imbalance is one of the major characteristics of OA[28]. Inflammatory cytokines including tumor necrosis factor (TNF)-α, interleukin (IL)-6, and IL-1β propagate the intra-articular inflammatory response and cartilage breakdown. To evaluate the role of *NUDT7* in inflammatory responses, we analyzed the alterations in the expression of mRNA of inflammatory cytokines and receptors by quantitative real-time RT-PCR (qRT-PCR) (Supplementary Fig. 7). Among them, the mRNA and protein levels of IL-1β were significantly increased both in human OA cartilage and the cartilage of DMM-induced *Nudt7*$^{-/-}$ mice (Fig. 4a,b). Overexpression of PGAM1 in iMACs of *Nudt7*$^{-/-}$ mice (Fig. 4c) significantly increased the transcript levels of pro-inflammatory cytokines, such as IL-1β, IL-6, and TNF. Moreover, introduction of PGAM1 displayed the typical characteristics of OA pathogenesis, i.e., an increased expression level of *MMP-13*; decreased expression levels of cartilage matrix genes such as *COL2A1*, *COMP*, and *ACAN* (Fig. 4h); increased expression and activation levels of cleaved caspase-3 and cleaved caspase-9 (Fig. 4d); stimulation of chondrocyte apoptosis

(Fig. 4e); and suppression of chondrocyte proliferation (Fig. 4f). Furthermore, we re-confirmed the increased expression levels of inflammatory cytokines in a dose-dependent manner by the introduction of PGAM1 (Fig. 4g).

**H3K4me3 is responsible for the activation of *PGAM1*.** H3K4me3 on a promoter is known to activate the gene expression. Here, we observed an increased level of H3K4me3 both in human OA cartilage compared with that in non-OA cartilage (Fig. 5a) and in the cartilage of *Nudt7*$^{-/-}$ mice compared with that in the cartilage of *Nudt7*$^{+/+}$ mice (Fig. 5b). Furthermore, the number of H3K4me3-positive cells increased in *Nudt7*$^{-/-}$ iMACs by approximately twofold (Fig. 5c); however, the number of H3K27me3-positive cells decreased (Supplementary Fig. 8). Next, we confirmed the direct interaction of H3K4me3 with the *PGAM1* promoter by a chromatin immunoprecipitation (ChIP) assay. As a result, we showed increased H3K4me3 levels in *Nudt7*$^{-/-}$ iMACs compared with those in *Nudt7*$^{+/+}$ (Fig. 5d).

The in vivo delivery of a *PGAM1*-specific shRNA (shPGAM1) into the cartilage of *Nudt7*$^{-/-}$ mice suppressed cartilage

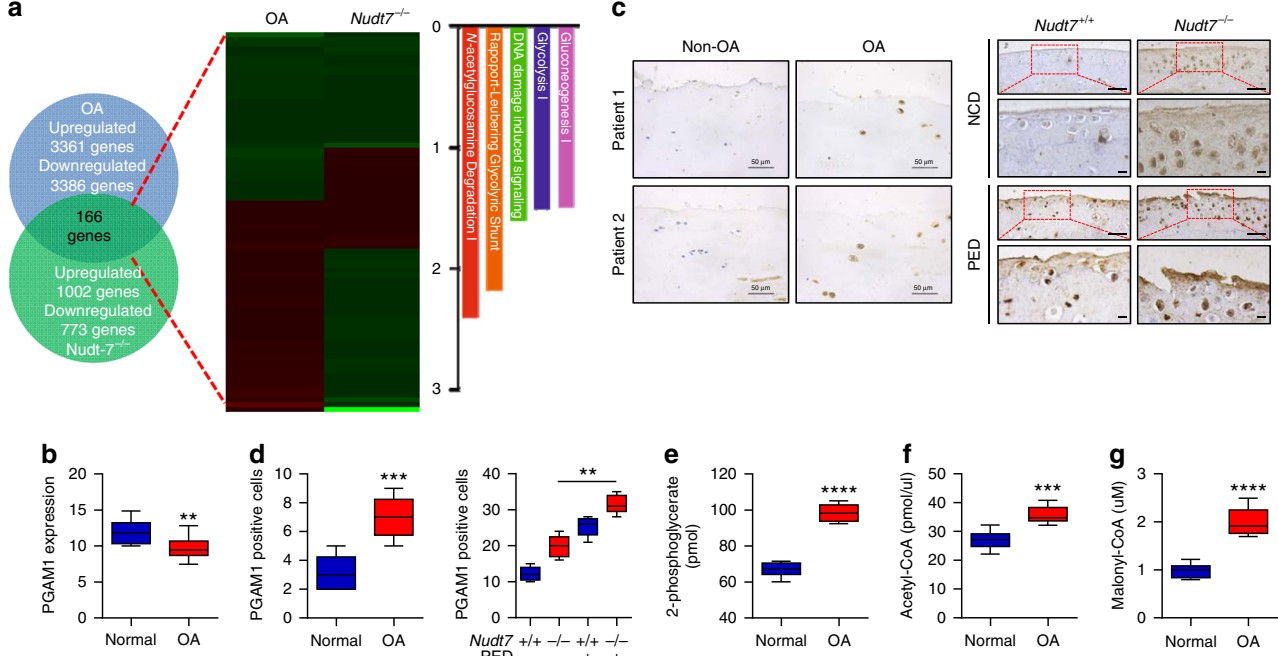

**Fig. 3** *NUDT7* deficiency affects glycolysis via upregulation of phosphoglycerate mutase 1 (PGAM1) expression during osteoarthritis (OA) pathogenesis. **a** Comparison of RNA sequencing between human OA chondrocyte and cartilage of *Nudt7⁻/⁻* mice, and possible pathological signaling pathways using Ingenuity Pathway Analysis (IPA). **b** Transcription level (ΔCt value) of *PGAM1* in human OA chondrocytes ($n = 15$) compared with that in normal chondrocytes ($n = 8$). **c** Immunohistochemistry of PGAM1 in human OA cartilages ($n = 5$ per group) and cartilages of *Nudt7⁻/⁻* mice ($n = 5$ per group) under NCD or PED conditions. Scale bars, 50 μm. **d** Positive cell number of PGAM1 in human OA cartilages ($n = 5$ per group) and cartilages of *Nudt7⁻/⁻* mice ($n = 5$ per group) under NCD or PED conditions from Fig. 3c. **e–g** Analysis of 2-phosphoglycerate, acetyl-CoA, and malonyl-CoA in immature murine articular chondrocytes (iMACs) of *Nudt7⁻/⁻* mice at postnatal day 6 ($n = 6$ per group). Values are means + s.d. An unpaired Student's *t* test was used for statistical analysis. **$P < 0.01$, ***$P < 0.001$, ****$P < 0.0001$.

degradation and lipid accumulation significantly (Fig. 6a and Supplementary Fig. 9). Moreover, co-introduction of *NUDT7* into mouse cartilage or human OA chondrocytes reversed the cartilage degradation and lipid accumulation induced by *PGAM1* over-expression (Fig. 6a,h, Supplementary Fig. 9). The in vivo delivery of *shPGAM1* into the cartilage of DMM-induced *Nudt7⁺/⁺* mice also suppressed cartilage degradation, whereas introduction of *PGAM1* induced severe cartilage degradation (Supplementary Fig. 10). The in vitro introduction of *shPGAM1* into iMACS of *Nudt7⁻/⁻* mice in phytol-enriched environment significantly inhibited lipid accumulation and the expression level of IL-1β (Fig. 6d). Accumulation of 2-PG, malonyl-CoA, and chondrocyte apoptosis in iMACs of *Nudt7⁻/⁻* mice were significantly reduced by the introduction of *shPGAM1* or *NUDT7* restoration (Fig. 6e–g).

## Discussion

Even though several factors, such as age, gender, and obesity are known to be involved in OA pathogenesis, the underlying regulatory mechanism is not well established. Recently, our laboratory reported dysfunction of cellular organelles, particularly peroxisomal dysfunction in OA chondrocytes, indicating the possible involvement of lipid homeostasis in OA pathogenesis. In fact, an increased level of very long chain fatty acids (VLCFA) was observed in OA chondrocytes. In the present study, we suggested that NUDT7 is one of the regulators responsible for maintaining cartilage homeostasis. To date, few studies have reported the pathological and biological role of NUDT7. NUDT7 regulates heme biosynthesis in pig skeletal muscle[27,29] and the NAD⁺/NADH balance in *Arabidopsis*[17]. In the present study, we found that the expression level of NUDT7 was significantly decreased in OA patients and modulation of *NUDT7* level altered

cartilage homeostasis by regulating chondrocyte survival and proliferation through alteration of lipid homeostasis. To evaluate the role of *NUDT7* during OA pathogenesis, *NUDT7* was globally deleted in mice. In *Nudt7⁻/⁻* mice, we found that hypertrophy zone was significantly decreased. However, limb length of post-natal mice and bone development, i.e., bone volume, trabecular bone thickness, trabecular bone number, and trabecular bone space, were not affected by *NUDT7* deficiency (Supplementary Fig. 11).

PGAM1 is a glycolytic enzyme that converts glucose to pyruvate by changing 3-phosphoglycerate (3-PG) to 2-phosphoglycerate (2-PG). PGAM1 plays an important role in several biological responses, including energy production and nucleotide biosynthesis[30,31]. The activity of PCAM1 is upregulated in several tumors, such as lung, colon, liver, breast, and leukemia[32,33], by regulating cell proliferation and migration. Knockdown of *PGAM1* inhibited cell proliferation and promoted cell apoptosis via S-phase cell cycle arrest in glioma[34]. Despite our understanding of how PGAM1 regulates glycolysis, rather less is known about the regulatory mechanism of PGAM1. In the present study, we found that transcription of *PCAM1* is regulated by *NUDT7*. In *NUDT7*-deficient conditions, significantly increased levels of *PGAM1*, as well as accumulation of acetyl-CoA were observed. Pyruvate synthesized by glycolytic enzymes is transported to mitochondria and is used for fatty acid synthesis; therefore, this increased acetyl-CoA level induced by *PGAM1* could be responsible for the accumulation of fatty acids in articular chondrocytes.

Gene expression could be controlled through multiple histone modifications, such as methylation and/or acetylation. Among them, histone 3 lysine specific trimethylation (such as H3K4me3, H3K9me3, H3K27me3, and H3K36me3) regulates gene

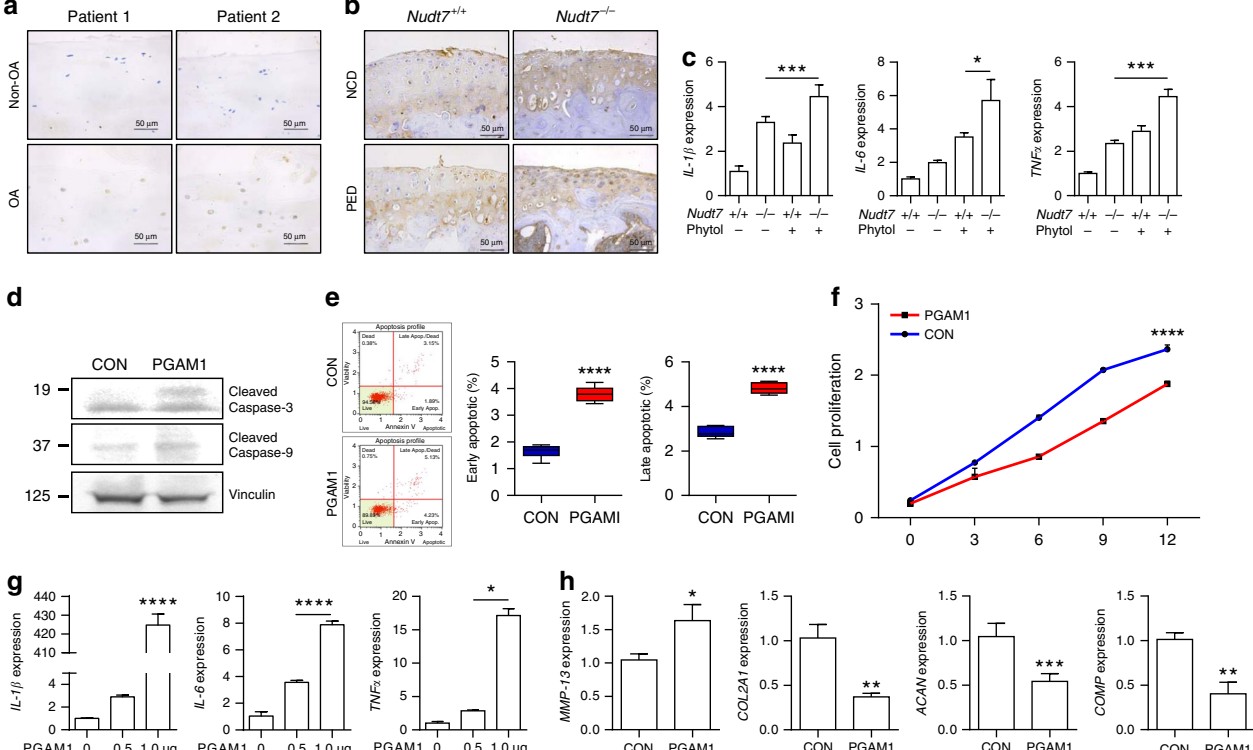

**Fig. 4** Upregulation of phosphoglycerate mutase 1 (*PGAM1*) expression induces pro-inflammatory cytokines. **a** Immunostaining of interleukin (IL)-1β in human OA chondrocyte compared with non-OA chondrocyte (*n* = 5 per group). **b** Immunostaining of IL-1β cartilage of DMM-induced *Nudt7⁻/⁻* mice (*n* = 5 per group). **c** Transcription levels of *IL-1β*, *-6*, and *TNFα* in iMACs of *Nudt7⁻/⁻* mice at postnatal day 6 under CON and phytol conditions (*n* = 3 per group). **d** Immunoblotting of cleaved caspase-3 and -9 with the introduction of PGAM1 into iMACs of *Nudt7⁺/⁺* mice. **e, f** Analysis of apoptotic cell death and cell proliferation with the introduction of *PGAM1* into iMACs of *Nudt7⁺/⁺* mice (*n* = 6 per group). **g** Transcriptional level of *IL-1β*, *-6*, and *TNFα* in iMACs of *Nudt7⁺/⁺* mice at postnatal day 6 transfected with *PGAM1* dose dependent (*n* = 3 per group). **h** Transcriptional levels of *MMP-13*, *COL2A1*, *ACAN*, and *COMP* transfected with *PGAM1*. Scale bars, 50 μm. Values are means + s.d. An unpaired Student's *t* test was used for statistical analysis. **\*\*P* < 0.01, \*\*\*P* < 0.001, \*\*\*\*P* < 0.0001

expression. H3K4me3 is an activating marker of gene expression that is localized in active chromatin[35,36], whereas H3K9me3 and H3K27me3 are repressive markers[37,38]. Recently, several reports demonstrated the involvement of H3K4me3 in degenerative diseases or cancers. Broad distributions of H3K4me3 on *DLX1*, *ELFN1*, *GAD1*, or *IGSF9B* in the human brain are known to be associated with neuronal functions and synaptic signaling[39]. H3K4me3 is involved in invasive metastasis in the MDA-MB-231 breast adenocarcinoma cell line[40] and enhances the expression of the cancer-related gene, *YAP1*, through the Menin and MLL complex in hepatocellular carcinoma[41]. In human osteoarthritic chondrocytes, a pro-inflammatory cytokine, IL-1β, induces H3K4me3 on the *iNOS* and *COX-2* gene promoters[42], and IL-6 induces H3K4me3 on *MMP-1*, *-3*, and *-9* promoters[43]. In the present study, we also observed that significantly increased level of H3K4me3 on the *PGAM1* promoter due to *NUDT7* deficiency was closely implicated in the pathogenesis of OA.

During the pathogenesis of OA, an increase in the levels of pro-inflammatory cytokines, such as IL-1β and TNF, was stimulated by catabolic actions in cartilage[5]. Recent reports indicated that a glycolytic enzyme, pyruvate kinase M2 (PKM2), activated inflammatory responses in the intestine[44] and coronary artery[45]. TNF-α-induced PKM2 activates apoptosis of intestinal epithelial cells by stimulations of BCL-XL, active caspase-3, and PARP[46], and promotes tumorigenesis by regulating the Warburg effect[47]. Our results suggested a close interaction between another glycolytic enzyme, PGAM1 and IL-1β, a well-known inflammation cytokine in OA pathogenesis. Consistent with the observations

reported in previous studies suggesting the involvement of inflammatory cytokines in the activation of MMP-13 in cartilage[5,48], we observed that an increased level of IL-1β, induced by *NUDT7* deficiency or *PGAM1* stimulation, resulted in the activation of MMP-13. Moreover, treatment of palmitic acid and linoleic acid to OA chondrocytes increased the level of IL-1β and MMP-13, and these increases in the levels of IL-1β and MMP-13 were reduced by co-introduction of *NUDT7* (Supplementary Fig. 12). These data suggest the possibility that the inflammatory actions of *NUDT7* might be the consequence of abnormal lipid deposition-related IL-1β expression.

In summary, our study suggested that peroxisomal dysfunction induced by NUDT7 deficiency affects glycolysis via the upregulation of PGAM1 expression. Dysregulation of NUDT77:PGAM1: IL-1β axis could be one of the key regulatory mechanisms in OA pathogenesis.

## Methods

**Experimental animals.** Destabilization of the medial meniscus (DMM) surgery was performed on the left knee joint of 8-week-old mice without cutting the ligament as a control. At 8 weeks after DMM surgery, the knee joint tissues were processed for histological analysis and scored for cartilage destruction using Mankin method.

**Generation of *Nudt7*-knockout mice.** *Nudt7*-specific TALEN mRNAs (50 ng/μl) were injected into the cytoplasm of C57BL/6 N mouse eggs and transferred into the oviducts of pseudo-pregnant foster female mice. T7E1 assays were performed using the genomic DNA samples from F₀ mutant mice. For routine PCR genotyping of F₁ progeny from selected F₀ founder mice, the following primer pair was designed to

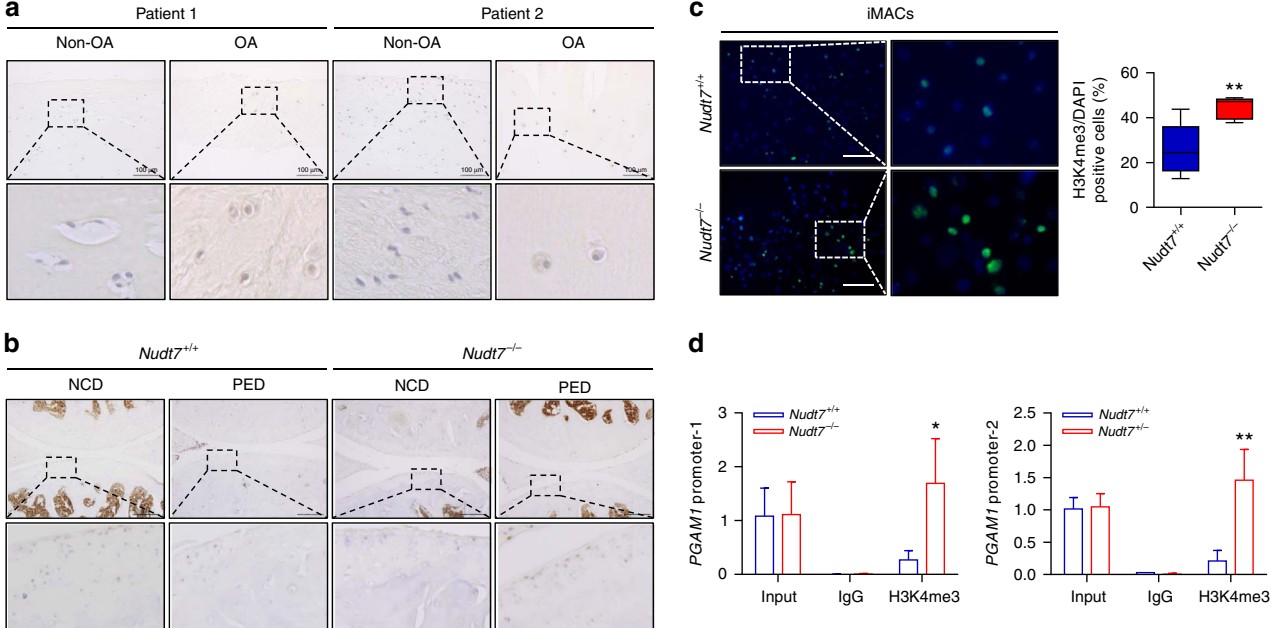

**Fig. 5** Histone 3 lysine trimethylation (H3K4me3) is involved in the transcriptional activation of phosphoglycerate mutase 1 (*PGAM1*). **a**, **b** Immunohistochemistry of H3K4me3 in human OA cartilage ($n = 5$ per group; scale bars, 100 μm) and cartilage of *Nudt7$^{-/-}$* mice ($n = 5$ per group; scale bars, 200 μm). **c** Immunocytochemistry of H3K4me3 and counting of H3K4me3-positive cells in immature murine articular chondrocytes (iMACs) of *Nudt7$^{-/-}$* mice ($n = 6$ per group; scale bars, 100 μm). **d** H3K4me3 binding assay on the *PGAM1* promoter using a chromatin immunoprecipitation (ChIP) assay ($n = 3$ per group). Values are means + s.d. An unpaired Student's *t* test was used for statistical analysis. *$P < 0.05$, **$P < 0.01$, ***$P < 0.001$, ****$P < 0.0001$

amplify a 209 bp PCR product from wild-type (WT) mice and TALEN-induced mutant alleles: forward 5′-CTGGCCAGAGGAGGAAAGTT-3′ and reverse 5′-ATGATCCTCTCCCTCCGACAT-3′.

**Immature mice articular chondrocytes culture**. Primary mouse articular chondrocytes were isolated from postnatal day 5 to 6 mice by dissection of the tibial plateaus and femoral condyles. Carefully peeled cartilage was digested with 3 mg/ml of collagenase D (Roche, 11 088 858 001) solution for 45 min and transferred to a petri dish containing 0.5 mg$^{-1}$ ml collagenase D solution and incubated overnight at 37 °C. After dispensing the digested cartilage through a 70-μm cell strainer, primary chondrocytes were cultured with Dulbecco's modified Eagle's medium (DMEM; Gibco-Invitrogen) supplemented with 10% fetal bovine serum (FBS; Gibco-Invitrogen), 100 units/ml of penicillin, and 100 units/ml of streptomycin at 37 °C in the presence of 5% CO$_2$ for 5 days.

**Human articular chondrocyte and chondrocyte cell line culture**. Human articular cartilage specimens were obtained from patients undergoing TKR and were designated as OA chondrocytes; normal chondrocytes were obtained from the biopsy samples of normal cartilage. Chondrocytes were extracted with collagenase and seeded at $1.5 \times 10^{-4}$ cells$^{-1}$ cm$^{-2}$ in DMEM supplemented with 10% FBS, 100 units$^{-1}$ ml penicillin, and 100 units$^{-1}$ ml streptomycin. Human OA chondrocytes were purchased from Cell Applications (402OAK-05a). The cells were maintained in the chondrocyte growth medium (Cell Applications, 411–500).

**Lentiviral constructs packaging and delivery**. *PGAM1, NUDT7, shPGAM1,* and *shNudt7*-containing lentiviruses were produced using the 3rd Generation Packaging Mix from the Applied Biological Materials, Inc. (ABM, LV053). Briefly, plasmids were transfected into Lenti-X 293 T cells (Clontech, 632180) using Lentifectin (ABM, G074) in OPTI-MEM I medium (ThermoFisher Scientific, 31985062) and cultured overnight. The supernatant was collected and lentiviral particles were concentrated using a Lenti-X Concentrator (Clontech, 631232) and stored at −80 °C. Cells were infected with lentivirus supernatant for 2 days in a humidified incubator at 37 °C in the presence of 5% CO$_2$. For in vivo articular cartilage delivery, concentrated lentivirus supernatant ($1 \times 10^9$ pfu) was injected into the intra-articular joint cavity of 8-week-old male mice once a week for 10 weeks.

**Histological staining**. For histological analysis, cartilage was fixed with 10% neutral buffered formalin for 1 day and decalcified in Calci-Clear Rapid (National Diagnostics, HS-105) for 36 h. After embedding in paraffin, sections were cut at 5-μm thickness at the weight baring part of joint and stained with safranin O and

scored according to the Mankin score. For immunohistochemistry, sections were incubated with primary antibodies overnight at 4 °C in a humidified chamber. Sections were subsequently developed using REAL EnVision Detection system (Dako, K5007) or an MOM kit (Vector Laboratories, BMK-2202).

**Antibodies**. The following antibodies were used in this study; rabbit anti-Collagen C1–2C (IBEX Pharmaceuticals, Cat #50–1035, 1:100), rabbit anti-cleaved Caspase-3 (Cell Signaling Technology, Cat #9664, 1:1000), rabbit anti-cleaved Caspase-9 (Cell Signaling Technology, Cat #9509, 1:1000), rabbit anti-PMP70 (Abcam, Cat #ab3421, 1:500), rabbit anti-FASN (Abcam, Cat #ab22759, 1:200), rabbit anti-H3K4me3 (Abcam, Cat #ab8580, 1:250), mouse anti-H3K27me3 (Abcam, Cat #ab6002, 1:250), rabbit anti-MMP13 (Biovision, Cat #3533, 1:200), rabbit anti-Aggrecan Neoepitope (NOVUS, Cat #100–74350), anti-human NUDT7 (Mybiosource, Cat #MBS416528), rabbit anti-PGAM1 (LSbio, Cat #LS-C143463, 1:200), rabbit anti-vinculin (Invitrogen, Cat #PA5-29688, 1:1000), customized rabbit anti-mouse NUDT7 (Younginfrontier, Korea), HRP-conjugated goat anti-rabbit IgG (Cat #ADI-SAB-300-J, Enzo Lifesciences), and goat anti-mouse IgG polyclonal antibodies (Cat # #ADI-SAB-100-J, Enzo Lifesciences).

**Immunofluorescence staining**. Cells grown on coverslips were fixed with 4% paraformaldehyde (PFA; Sigma-Aldrich, 441244) for 10 min and permeabilized with 0.1% Triton-X-100 solution for 5 min at room temperature. After washing three times in ice-cold 1 × phosphate-buffered saline (PBS), cells were blocked with 3% normal goat serum (NGS; Vector Laboratory, S-1000) and incubated with primary antibodies and fluorescence-conjugated secondary antibodies. Nuclei were stained with 4,6-diamidino-2-phenylindole(DAPI)-containing mounting medium (Vector Laboratory, S-1200).

**Western blotting**. For western blotting analyses, total protein samples were isolated using radioimmunoprecipitation assay (RIPA) lysis buffer (65 mM Tris-HCl, pH 7.5, 150 mM NaCl, 1 mM EDTA, 1% Nonidet P-40, 0.5% sodium deoxycholate, and 0.1% SDS, protease inhibitor cocktail, and phosphatase inhibitor). The BCA Protein Assay Kit (ThermoFisher Scientific, 23225) was used to determine the protein concentrations. Proteins were separated using 10% SDS-PAGE gels and then transferred to nitrocellulose membranes. After the membranes were blocked in 5% skim milk, they were incubated overnight at 4 °C with primary antibodies and then for 1 h at room temperature with the horseradish peroxidase-conjugated corresponding secondary antibodies. The reacted proteins were detected and visualized using either an electrochemiluminescence (ECL) system (ThermoFisher Scientific, 32106) or chemidoc (Fujifilm Life Science, LAS 4000). Uncropped scan of original blots are shown in Supplementary Fig. 13.

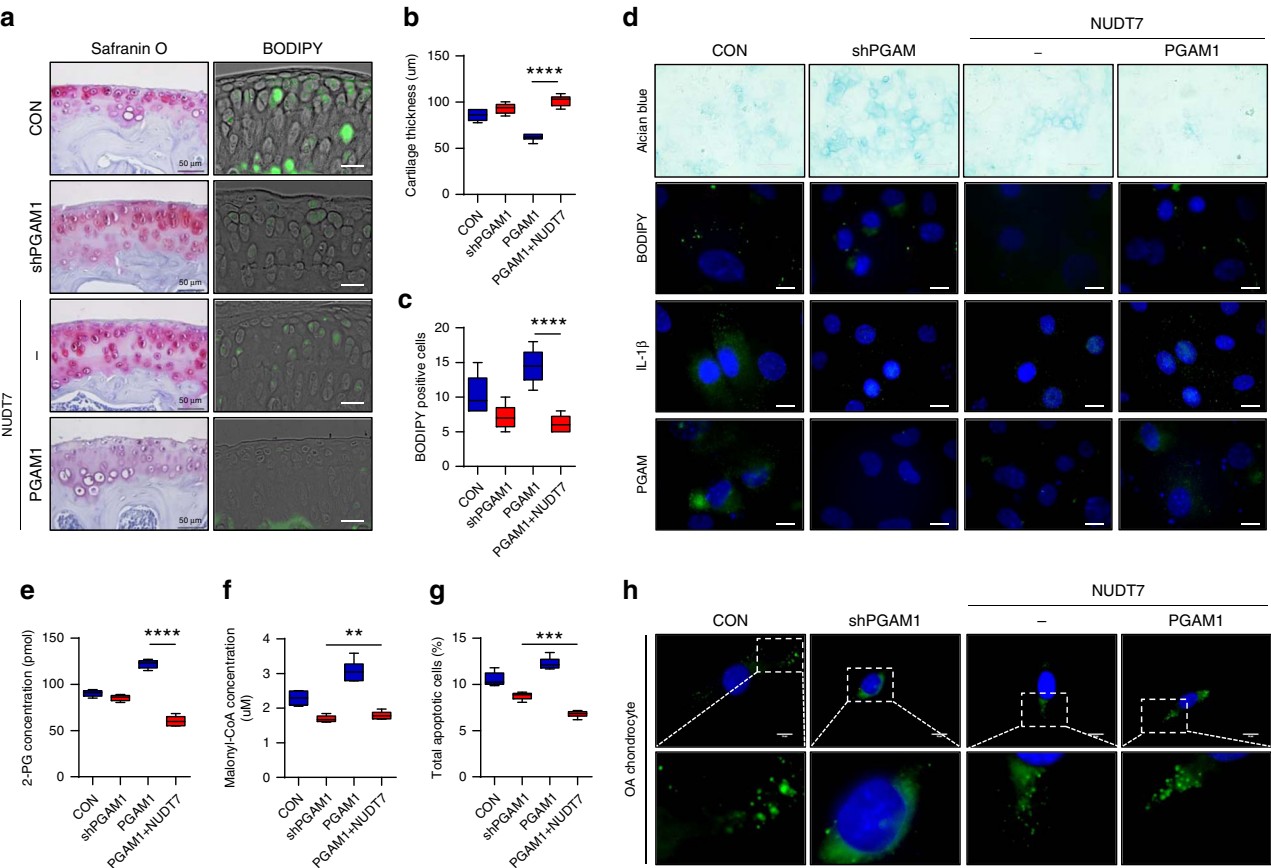

**Fig. 6** Cartilage destruction by phosphoglycerate mutase 1 (*PGAM1*) is reversed by co-introduction of *NUDT7*. **a** Staining with safranin O and BODIPY$^{493/}$ $^{508}$ using the cartilage of *Nudt7$^{-/-}$* mice infected with lentivirus containing either *PGAM1* shRNA (*shPGAM*) or *NUDT7* into the joint cavity 10 weeks after destabilization of the medial meniscus (DMM) surgery (*n* = 5, Scale bar, 50 μm). **b**, **c** Analysis of cartilage thickness, BODIPY$^{493/508}$-positive cells from staining with safranin O, and BODIPY$^{493/508}$ from Fig. 6a. **d** Staining of BODIPY$^{493/508}$, IL-1β, and PGAM1 with the introduction of *shPGAM1* or *NUDT7* and/or *PGAM1* into iMACs of *Nudt7$^{+/+}$* mice (*n* = 6 per group). Scale bars, 10 μm. **e–g** Analysis of 2-phosphoglycerate, malonyl-CoA, and apoptotic cell death in immature murine articular chondrocytes (iMACs) of *Nudt7$^{-/-}$* mice infected with lentivirus containing *shPGAM* or *NUDT7* and/or *PGAM1* (*n* = 6 per group). **h** BODIPY$^{493/508}$ staining of human osteoarthritis (OA) chondrocyte infected with lentivirus containing *shPGAM* or *NUDT7* and/or *PGAM1* (*n* = 6 per group). Scale bars, 10 μm. Values are means + s.d. An unpaired Student's *t* test was used for statistical analysis. \*\**P* < 0.01, \*\*\**P* < 0.001, \*\*\*\**P* < 0.0001

**qRT-PCR**. RNA was isolated using RNAisoplus (Takara, 9109) and reverse transcribed using 5X All-In-One RT MasterMix (ABM, G492). Quantitative PCR was performed using 2X SYBR-Green MasterMix (Geneer, Q903L). The qRT-PCR primer sequence used in this study are listed in Supplementary Table 1 and Supplementary Table 2. RN18S was used as endogenous control, and the expression levels were analyzed and visualized using PermutMatrix version 1.9.3.

**Catalase and GPx assay**. Catalase activity and GPx activity were measured using their respective assay kits (BioVision, K773-100, K762-100) according to the manufacturer's instructions. The enzyme activity was calculated as nanomoles of GSH$^{-1}$ min$^{-1}$ mg of protein.

**Apoptotic cell death and cell proliferation assay**. Apoptotic cell death was assessed using an Annexin V & Dead Cell Assay Kit (EMD Merck Millipore, MCH100105) with the Muse Cell Analyzer (EMD Merck Millipore). Cell proliferation was assessed using the Quick Cell Proliferation Colorimetric Assay Kit (BioVision, K301-500) according to the manufacturer's instructions.

**Microarray**. Total RNA was extracted using a RecoverAll Total Nucleic Acid Isolation (ThermoFisher Scientific #AM1975) from FFPE cartilage samples of *Nudt7$^{+/+}$* and *Nudt7$^{-/-}$* mice. Whole-transcript expression array analysis was performed according to the Affymetrix GeneChip Whole Transcript PLUS reagent method. cDNA was synthesized using a GeneChip Whole Transcript Amplification kit. Sense cDNA was then fragmented, biotin-labeled with TdT using the GeneChip WT Terminal labeling kit, and hybridized to the Affymetrix GeneChip Mouse 2.0 ST Array for 16 h at 45 °C. The hybridized array was washed and stained on a GeneChip Fluidics Station 450 and scanned on a G GCS3000 Scanner (Affymetrix).

Signal values were computed using the Affymetrix GeneChipCommand Console software.

**RNA sequencing**. RNA sequencing performed using an Illumina HiSeq4000 instrument and libraries were quantified using quantitative real-time PCR (qPCR), and their quality was checked using an Agilent Technologies 2100 Bioanalyzer. The raw data were calculated as fragments per kilobase of transcript per million mapped reads (FPKM) of each sample using the cufflinks software. Data was transformed logarithmically and normalized using the quantile normalization method.

**ChIP assay**. The H3K4me3-ChIP assay was performed using a ChIP-IT Express Kit (Active Motif, 53035) as previously described[49]. The genomic DNA was sheared using components of the ChIP-IT Express Kit (Active Motif, 53035), and 20 μg of genomic DNA was immunoprecipitated with 3 μg of H3K4me3 (Abcam, ab8580) antibody and Protein G-coupled magnetic beads (Active Motif, 53014). The bead-bound chromatin was eluted with 1% SDS and 0.1 M NaHCO$_3$, and purified using phenol–chloroform. The *PGAM1* promoter sequences for qRT-PCR were obtained from the Eukaryotic Promoter Database (http://epd.vital-it.ch), and the *ACTB* gene was used for endogenous normalization.

**2-Phosphoglycerate assay**. The 2-PG level was measured using 2-Phosphoglycerate Colorimetric/Fluorometric Assay Kit (BioVision, K778-100) according to the manufacturer's instructions.

**Acetyl-CoA and malonyl-CoA assay**. The acetyl-CoA concentration of each sample was measured using a PicoProbe acetyl-CoA assay kit (Abcam, ab87546).

The fluorescent signal was measured at excitation/emission wavelengths = 535/589 nm using a Spectra Max M3 instrument (Molecular Devices, Sunnyvale, CA). The malonyl-CoA concentration of each sample was measured using a Mouse Malonyl coenzyme A ELISA Kit (MyBioSource, MBS705127).

**Inflammatory cytokine expression**. Total RNA was extracted using RecoverAll Total Nucleic Acid Isolation (ThermoFisher Scientific, AM1975) from the FFPE cartilage samples, and gene expression was analyzed using RT$^2$ Profiler PCR Array Mouse Inflammatory Cytokines & Receptors (Qiagen, PAMM-011). Briefly, RNA was reverse transcribed using the RT$^2$ First Strand Kit (Qiagen, 330401), and PCR reaction was carried out using the QuantStudio 6 K Flex 384-well instrument (Applied Biosystems). The mRNA expression levels were analyzed using the $\Delta$ cycle threshold (Ct) method ($2^{-\Delta\Delta Ct}$).

**Ethical approval**. All studies were performed following the approval from the Wonkwang University Animal Care and Use Committee and were in compliance with the institutional guidelines (#WKU18-23). Human cartilage tissue collection was approved by the Human Subjects Committee of Wonkwang University Hospital and studies were performed in compliance with the institutional guidelines (WKUH 201605-HRBR-041). Written informed consent was obtained from all adult patients or at least one guardian of each patient prior to the start of the experiment.

**Data availability**. The data sets generated during and/or analyzed during the current study are available from the corresponding author on reasonable request.

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

## Acknowledgements
This work was supported by the National Research Foundation of Korea (NRF) grant funded by the Korea government (MSIP) [2016R1A2B4010577 and 2016R1D1A1B03930955].

## Author contributions
Conceptualization: E.-J.J., Data curation: J.S., Formal analysis: J.S., Funding acquisition: J.S. and E.-J.J., Investigation: J.S., I.-J.B., and E.-J.J., Resources: I.-J.B. and C.-H.C., Writing original draft: E.-J.J.

## Additional information

**Competing interests:** The authors declare no competing interests.

