## [Peer Review File · Nature Communications]

Reviewers' comments:

Reviewer #1 (Remarks to the Author):

This is a very interesting and novel study where the Authors make strong conclusions following a multi-pronged approach to challenge the main hypothesis. The series of experiments conducted is logical and demonstrates a deep knowledge of the field. As such this manuscript presents a continuation of previous work with a link between metabolic status and production of IL-1 in chondrocytes. I have enjoyed reading the manuscript and I have the following comments:

1. A key experiment missing is testing whether overexpression of NUDT7 in OA chondrocytes prevents apoptosis and lipid accumulation.
2. I cant find anywhere the methods for how the Authors selected slides for analysis of the DMM. They only show one slide with only the cartilage visible so one cannot tell which part of the joint is being analysed. As DMM is a mechanically induced disease, Authors must do analysis in the weight bearing part of the joint (within 100um either side of slide with the smallest meniscus slice). This is also important for assessing cartilage thickness. They might want to add an OARSI score of the cartilage to assess OA.
3. Some of the safranin O stainings require some improvement. Particularly Fig 1a - wrong colour, should be red/orange and not much change in colour intensity between preserved and OA tissue pieces. Also Fig 6a - figure not really clear, looks like nud7 overexpression decreases staining and PGAM1 over expression improves cartilage health? while the opposite is being described.
4. Fig1d is also unclear, why does shNud7a make nud7 expression go up? which isoform do they assess and which shRNA did they take forward? what is the contribution of the other isoforms?
5. Would be nice to show the accumulation of fatty acids in the chondrocytes as the Authors hint at in the discussion.

Reviewer #2 (Remarks to the Author):

This paper reports on the role of the peroxisomal nudix hydrolase (NUDT7) in cartilage and osteoarthritis (OA). Results show that is reduced in human OA chondrocytes. Knock down of NUDT1 impairs lipid homeostasis and Nudt7^{-/-} mice displayed more severe OA. This was associated with accumulation of lipids via peroxisomal dysfunction, increased IL1b expression and chondrocyte apoptosis.

Nudt7 knockout affected the glycolytic pathway and identified Pgam1 as one of significantly altered genes. Overexpression of PGAM1 in chondrocytes induced the accumulation of lipids upregulation of IL-1 β expression, and apoptotic cell death.

Negative actions of PGAM1 in maintaining cartilage homeostasis were reversed by the co-introduction Nudt7.

The novelty of this paper is that it is the first to investigate a nudix hydrolase in cartilage and OA.

A main limitation of the study is that mice with global deletion of Nudt7 were used. This needs to at least be addressed in the discussion,

Information needs to be provided whether the Nudt7 KO mice had normal skeletal development and normal joint growth and maturation until the time when DMM surgery was performed.

Authors also need to address whether Nudt7 deficiency affects other joint tissues (synovium, bone). The interpretation of the data from the KO mice is too narrowly focused on cartilage.

The discussion should also address whether the effects on inflammation are a direct effect of Nudt7 and/or a consequence of abnormal lipid deposition-related IL-1 expression.

The results in Fig1 C should be shown in a different format so that the OA values are represented as being lower.

Page 6, line 126: explain the purpose of feeding a phytol-enriched diet

Methods need to explain dosage and frequency of intraarticular shRNA injections. How were dosages determined and validated?

Reviewer #1

1) A key experiment missing is testing whether overexpression of NUDT7 in OA chondrocytes prevents apoptosis and lipid accumulation.

We appreciate the insightful comment. We showed the result in supplementary figure 2. Also please see attached following figure for reviewers.

Apoptosis analysis (For reviewers)

Lipid accumulation in OA chondrocytes by modulating Nudt-7 (For reviewers)

	CON	Nudt-7	shNudt-7	Nudt-7 + shNudt-7
Live	91.20	93.95	89.20	91.35
Early Apop	7.40	5.70	9.90	8.00
Late Apop	1.40	0.30	0.85	0.65
Total Apop	8.80	6.00	10.75	8.65

2) I can't find anywhere the methods for how the authors selected slides for analysis of the DMM. They only show one slide with only the cartilage visible so one cannot tell which part of the joint is being analysed. As DMM is a mechanically induced disease, Authors must do analysis in the weight bearing part of the joint (within 100um either side of slide with the smallest meniscus slice). This is also important for assessing cartilage thickness. They might want to add an OARSI score of the cartilage to assess OA.

(1) When we embedded the cartilage (joint) with same angles for every sample, sectioned, and choose section at the site of weight bearing part of joint. Please find attached following figure for reviewer.

Sorry for missing for this information and we added that we used the sections at the site of weight bearing part of joint in the method.

(2) We followed this suggestion and added an OARSI score of the cartilage in figure 2b.

3) Some of the safranin O stainings require some improvement. Particularly Fig 1a – wrong colour, should be red/orange and not much change in colour intensity between preserved and OA tissue pieces. Also Fig 6a – figure not really clear, looks like nudt7 overexpression decreases staining and PGAM1 over expression improves cartilage health? While the opposite is being described.

We followed this suggestion and changed the safranin O staining of figure 1a and figure 6a as well as supplementary figure 11.

4. Fig 1d is also unclear, why does shNudt7a make nudt7 expression go up? Which isoform do they assess and which shRNA did they take forward? What is the contribution of the other isoforms?

We had screened three independent shRNAs against Nudt7 and used two shRNAs which showed the suppression of Nudt7 in this study.

5) Would be nice to show the accumulation of fatty acids in the chondrocytes as the Authors hint at in the discussion

We had assayed the level of free fatty acid in the chondrocytes and added in supplementary figure 1f.

As well, we have reported previously that VLCFA and LCFA were accumulated in OA chondrocytes (Song J, Kang YH, Yoon S, Chun CH, Jin EJ. HIF-1 α :CRAT:miR-144-3p axis dysregulation promotes osteoarthritis chondrocyte apoptosis and VLCFA accumulation. *Oncotarget*. 2017 Sep 1;8(41):69351-69361). Please see below attached figure 1 in this paper.

Figure 1: Lipid accumulation is involved in osteoarthritis pathogenesis. A. Chondrocytes isolated from either healthy (non-osteoarthritis), or osteoarthritis cartilage from patients who underwent total knee replacement surgery and had different BMIs, were stained with BODIPY. B. Chondrocytes (normal, non-osteoarthritis, and osteoarthritis) were isolated, and total lipid content was analyzed using gas chromatography/mass spectrometry. Lipids were then separated into VLCFA, and either long or medium chain fatty acids (LCFA and MCFAs, respectively). The data shown are representative of independent data from at least four patients.

Reviewer #2

1) A main limitation of the study is that mice with global deletion of Nudt7 were used. This needs to at least be addressed in the discussion.

We followed this suggestion and addressed this matter in the discussion

2) Information needs to be provided whether the Nudt7 KO mice had normal skeletal development and normal joint growth and maturation until the time when DMM surgery was performed. Authors also need to address whether Nudt7 deficiency affects other joint tissues (synovium, bone). The interpretation of the data from the KO mice is too narrowly focused on cartilage.

We followed this suggestion and show the result in supplementary figure 12.

3) The discussion should also address whether the effects on inflammation are a direct effect of Nudt7 and/or a consequence of abnormal lipid deposition-related IL-1 expression.

We had performed the experiments with fatty acid treatment, analyzed the level of IL-1 \$\beta\$ and MMP13 and added in supplementary figure 12. Also, we followed this suggestion and addressed this matter in the discussion

4) The results in Fig 1c should be shown in a different format so that the OA values are represented as being lower.

We had represented dCt values and this time we had changed into fold change in revised manuscript.

5) Page 6, line 126: explain the purpose of feeding a phytol-enriched diet.

We followed this suggestion and addressed this matter in the text (page 6)

6) Methods need to explain dosage and frequency of intraarticular shRNA injections. How were dosages determined and validated?

We had addressed this matter in the method with reference (#51).

REVIEWERS' COMMENTS:

Reviewer #1 (Remarks to the Author):

I think the Authors have addressed my comments and I have not much else to ask. The data are well presented and support the conclusions put forward by the study.

Reviewer #2 (Remarks to the Author):

Authors addressed this reviewer's comments.

The methods section needs additional information about bone assessment.

REVIEWERS' COMMENTS:

Reviewer #1 (Remarks to the Author):

I think the Authors have addressed my comments and I have not much else to ask. The data are well presented and support the conclusions put forward by the study.

: We are truly thankful for the comment.

Reviewer #2 (Remarks to the Author):

Authors addressed this reviewer's comments.

The methods section needs additional information about bone assessment.

: We had added the information for bone assessment in Supplementary Method section.